# Functional Characterization of MIP_07528 of *Mycobacterium indicus pranii* for Tyrosine Phosphatase Activity Displays Sensitivity to Oxidative Inactivation and Plays a Role in Immunomodulation

**DOI:** 10.3390/biology14050565

**Published:** 2025-05-18

**Authors:** Raunak Raunak, Roopshali Rakshit, Aayush Bahl, Soumya Sinha, Saurabh Pandey, Sashi Kant, Deeksha Tripathi

**Affiliations:** 1Microbial Pathogenesis and Microbiome Lab, Department of Microbiology, School of Life Sciences, Central University of Rajasthan, Ajmer 305817, Rajasthan, India; raunak.mailbox@gmail.com (R.R.); roopshalirakshit@gmail.com (R.R.); aayush.rkbahl@gmail.com (A.B.); sinhasoumya101017d@gmail.com (S.S.); 2Department of Biochemistry, School of Chemical and Life Sciences, Jamia Hamdard, New Delhi 110062, Delhi, India; saurabhpandey@jamiahamdard.ac.in; 3Department of Immunology and Microbiology, University of Colorado School of Medicine, Anschutz Medical Campus, Aurora, CO 80045, USA

**Keywords:** *Mycobacterium Indicus Pranii*, *Mycobacterium tuberculosis*, tyrosine phosphatase, immunomodulation, host–pathogen interactions

## Abstract

*Mycobacterium indicus pranii*, originally developed as a vaccine for leprosy, has been found to possess significant immunomodulatory properties, yet the underlying molecular mechanisms are not well understood. This study aimed to investigate the role of a specific protein, MIP_07528, identified as a potential protein tyrosine phosphatase B ortholog. The research revealed that this protein is conserved among pathogenic mycobacteria and exhibits phosphatase activity, which helps the bacteria adapt to stress. Notably, MIP_07528 was shown to reduce inflammation in immune cells while promoting anti-inflammatory responses. These findings suggest that MIP_07528 plays a crucial role in the unique characteristics of *Mycobacterium indicus pranii*. Understanding these mechanisms provides valuable insights into mycobacterial interactions with the immune system and opens avenues for developing innovative strategies to treat infectious diseases. This research holds significant promise for advancing antimicrobial therapies, ultimately improving health outcomes in communities impacted by mycobacterial infections.

## 1. Introduction

*Mycobacterium Indicus Pranii* (MIP), previously known as *Mycobacterium w* (Mw), is a rapidly growing, cultivable, non-tuberculous mycobacterial species with unique immunomodulatory properties [1]. As an evolutionary progenitor of opportunistic pathogens within the *Mycobacterium avium* complex (MAC), MIP exhibits intermediary phenotypic and genetic characteristics that bridge saprophytic and pathogenic mycobacteria. Its shared epitopes with both *Mycobacterium leprae* and *Mycobacterium tuberculosis* (Mtb) enable cross-reactive immune responses [2]. MIP has been reported to stimulate both innate and adaptive immunity by activating macrophages and dendritic cells, leading to pro-inflammatory cytokine induction. Additionally, it promotes Th1 and Th17 responses while downregulating Th2 pathways, facilitating protective memory T-cell responses in the pulmonary airways [3]. Initially developed as a heat-inactivated vaccine against leprosy [4], MIP has shown therapeutic efficacy across diverse conditions, including non-small cell lung carcinoma [5,6,7,8], sepsis [9,10], warts [11], psoriasis [12], category II tuberculosis [13,14], HIV [15], and COVID-19 [16,17]. Despite its comprehensive immunomodulatory effects and therapeutic potential, the molecular mechanisms underlying MIP’s attenuated virulence remain inadequately characterized.

Phosphatases function as critical enzymatic regulators that orchestrate diverse cellular processes through the targeted dephosphorylation of phosphoproteins, phosphoinositides (PIs), and various other phosphorylated biomolecules. Recent studies demonstrated that bacterial phosphatases, such as PhpP in *S. pneumoniae*, serve as critical regulators of virulence and could be targeted for therapeutic interventions [18]. Phosphatases secreted by intracellular pathogens like *Mycobacterium tuberculosis* have been found to manipulate host PI metabolism, disrupt phagosome maturation, and interfere with immune signaling pathways, facilitating immune evasion and establishment of persistent infection. Protein tyrosine phosphatase B in *M. tuberculosis* (PtpB-Mtb) has been reported to be an established virulence factor with dual protein and lipid phosphatase activity and triple specificity, playing a key role in immunomodulation and creating a permissible intracellular niche for sustained mycobacterial persistence within host macrophages [19,20,21,22]. The enzymatic versatility of PtpB-Mtb allows it to simultaneously disrupt multiple host-signaling pathways essential for immune defense. Interestingly, PtpB-Mtb is characterized by a unique, protective two-helix lid covering its active site—an evolutionary adaptation specific to mycobacterial phosphatases [21]. Furthermore, the absence of a human ortholog, coupled with its secretory nature and crucial role in mycobacterial pathogenesis, PtpB-Mtb has emerged as a promising therapeutic target, with inhibitors showing efficacy in reducing mycobacterial burden in experimental models [23,24,25].

In light of these findings, the evolutionary relationship between pathogenic and non-pathogenic mycobacteria suggests that MIP may possess functionally similar phosphatases despite its attenuated virulence. MIP’s documented immunomodulatory effects indicate that its PtpB ortholog may play a crucial role in host-mycobacterium interactions. Understanding these mechanisms could illuminate how phosphatases influence immune signaling pathways and potentially explain MIP’s therapeutic efficacy across various clinical applications. Furthermore, structural and functional characterization of MIP PtpB enables identification of conserved catalytic elements while highlighting distinctions between pathogenic and non-pathogenic variants, facilitating development of selective inhibitors targeting pathogenic mycobacteria. Thus, it seems imperative to focus on putative phosphatases in order to devise antimycobacterial strategies as well as to determine the potential immunomodulation pathways. Considering the utility of MIP as a potent intervention against mycobacterial infections, there is a pressing need to investigate the roles of various MIP proteins—especially its phosphatases—to better understand their immunomodulatory roles and develop new antimycobacterial approaches.

## 2. Materials and Methods

### 2.1. In Silico Characterization of MIP_07528 as a Putative PtpB

To investigate the evolutionary conservation of MIP_07528, a series of in silico analyses were performed. The amino acid sequence of MIP_07528 from *Mycobacterium indicus pranii* was initially obtained from the NCBI database. Subsequently, nucleotide and protein sequence identity between MIP_07528 (AFS17053) and PtpB orthologs from *Mycobacterium tuberculosis* H37Rv (NP_214667), *Mycobacterium avium* (WP_009979776), *Mycobacterium bovis* (CAB5247863), and *Mycobacterium smegmatis* (ABK73175) were assessed. This was achieved using the BLASTn and BLASTp algorithms, respectively. The resulting percentage identity scores, reflecting evolutionary conservation, were then recorded. To further examine sequence conservation, a multiple sequence alignment of the aforementioned homologous sequences from the selected mycobacterial strains was conducted using ClustalW, EMBL- EBI. Subsequently, putative active sites for PtpB from each species were identified using the SiteMap module of Schrodinger. This identification was based on sequence homology and the well-established PtpB active site signatures. The three-dimensional structures of these identified active sites were then predicted. Visualization of the predicted structures was performed using PyMol 3.1 by Schrödinger. Conserved residues within the active sites were specifically highlighted to facilitate a detailed comparison of structural similarities. Phylogenetic analysis was performed using MEGA, version 11.0.13 (Molecular Evolutionary Genetics Analysis) to determine evolutionary relationships between the putative MIP_07528 and mycobacterial PtpB orthologs. The maximum likelihood method with 1000 bootstrap replicates was employed, with bootstrap values > 50% considered significant. PtpB protein sequences from the selected mycobacterial species were used for tree construction, and the *Nocardia farcinica* (NFA) sequence served as an outgroup.

### 2.2. Bacterial Strains and Culture Conditions

Recombinant protein expression utilized the *Escherichia coli* BL21 (DE3) strain (Novagen- United states). *Mycobacterium indicus pranii* (MIP) was cultivated in Middlebrook 7H9 broth enriched with 10% OADC and 0.05% Tween-80, under identical conditions as those established for *Mycobacterium smegmatis* mc^2^ 155 (ATCC 700084). The *M. smegmatis* strain was sourced directly from ATCC. For gene amplification purposes, genomic DNA from *Mycobacterium tuberculosis* H37Rv was graciously provided by Prof. Rakesh Bhatnagar’s laboratory at Jawaharlal Nehru University, New Delhi, India.

### 2.3. Assessing Phosphatase Activity in Mycobacterium Indicus Pranii

*Mycobacterium indicus pranii* (MIP) was cultured in Middlebrook 7H9 broth supplemented with 10% OADC enrichment and 0.05% Tween-80 at 37 °C until mid-log phase. Following incubation, cultures were subjected to oxidative and acidic stress conditions for 30 min. Oxidative stress was induced by exposure to varying concentrations of hydrogen peroxide (H_2_O_2_), specifically 31.25, 62.5, 125, 250, and 500 μM. Acidic stress was induced by culturing MIP in media adjusted to pH 5.5, while a pH of 7.5 served as the control.

After stress exposure, bacterial cells were harvested by centrifugation. Supernatants were collected and filtered through a 0.22 μm membrane to obtain culture filtrate. Cell pellets were washed with phosphate-buffered saline (PBS) and resuspended in 1x PBS. Cell lysis was performed by sonication, followed by centrifugation to remove cell debris.

Phosphatase activity in culture filtrates and cell lysates was assessed using *p*-nitrophenyl phosphate (pNPP) as a substrate. The reaction mixture, containing 10 μg of cell lysate or 10 μg of enzyme from culture filtrate and 10 mM pNPP in assay buffer (20 mM Tris, 150 mM NaCl, pH 7.5), was incubated at 37 °C for 30 min. The reaction was quenched by adding 0.4 M NaOH. The absorbance at 415 nm was measured using a spectrophotometer. Phosphatase activity was expressed as picomoles of *p*-nitrophenol produced per minute per milligram of protein.

Total RNA was extracted from MIP cultures under standard, oxidative (H_2_O_2_ concentrations: 31.25, 62.5, 125, 250, and 500 μM), and acidic stress (pH 5.5 and 7.5) conditions, for 30 min, using TRIzol reagent. The concentration and purity of RNA were determined using a BioTek Synergy HTX Multimode Reader- United States. cDNA synthesis was performed using a reverse transcription kit (Verso cDNA Synthesis Kit- Thermo Scientific, United states). Quantitative PCR was carried out using gene-specific primers and SYBR Green Master Mix on a real-time PCR system. Relative gene expression levels were calculated using the 2^−ΔΔCT^ method, with normalization to the housekeeping gene *sigA*. The amplified product was verified by sequencing and compared to the MIP_07528 gene sequence in the NCBI database.

### 2.4. Cloning and Expression of MIP_07528 and ptpB-Mtb in Heterologous Host

The open reading frames of MIP_07528 and Rv0153c (*ptpB*-Mtb) were amplified by PCR using genomic DNA from MIP and *M. tuberculosis* H37Rv, respectively. The primers used for amplification included *BamHI* and *HindIII* restriction sites (Table 1). The amplified PCR products and the pET28a expression vector were digested with *BamHI* and *HindIII* restriction enzymes (NEB), followed by ligation using T4 DNA ligase (NEB). The resulting recombinant constructs were transformed into *E. coli* DH5α competent cells. The integrity of these constructs was further confirmed by Sanger sequencing.

The recombinant pET28a constructs containing MIP_07528 or *ptpB*-Mtb were transformed into *E. coli* BL21 (DE3) expression cells. A single colony was inoculated into Luria–Bertani (LB) broth and grown at 37 °C until the optical density at 600 nm (OD600) reached 0.4. Protein expression was induced by adding isopropyl β-D-1-thiogalactopyranoside (IPTG) to a final concentration of 0.1 mM. The cultures were incubated at 20 °C for 20 h. The bacterial cells were harvested by centrifugation (6000 rpm, 15 min, 4 °C) and stored at −80 °C until purification.

The cell pellets were resuspended in lysis buffer (50 mM Tris, 250 mM NaCl, 5 μM β-mercaptoethanol, 5% glycerol) containing 1 mM phenylmethylsulfonyl fluoride (PMSF). Cells were lysed by sonication (10 s on/off cycles, 32 Hz, 15 min on ice). The lysates were centrifuged (6000 rpm, 30 min, 4 °C) to remove cellular debris. The supernatant was filtered through a 0.45 μm membrane and loaded onto a nickel-nitrilotriacetic acid (Ni-NTA) affinity chromatography column pre-equilibrated with lysis buffer. The column was washed with lysis buffer containing 30 mM imidazole to remove non-specifically bound proteins. Bound proteins were eluted using a linear gradient of imidazole (50–200 mM) in lysis buffer. Fractions containing the target protein, as determined by SDS-PAGE, were pooled and dialyzed against lysis buffer at 4 °C using a 10 kDa molecular weight cut-off (MWCO) dialysis membrane. The concentration of the purified protein was determined by the Bradford assay.

### 2.5. Validation of rMIP_07528 Using Anti-His Antibody

The purified recombinant proteins were analyzed by SDS-PAGE and transferred to a polyvinylidene difluoride (PVDF) membrane. The membrane was blocked with 5% non-fat milk in Tris-buffered saline with 0.1% Tween-20 (TBST) for 1 h at room temperature. The membrane was incubated with anti-histidine tag antibody (1:5000 dilution) overnight at 4 °C, followed by incubation with a horseradish peroxidase (HRP)-conjugated secondary antibody (1:10,000 dilution) for 1 h at room temperature. The presence of the His-tagged MIP_07528 and PtpB-Mtb proteins was visualized using enhanced chemiluminescence (ECL) and imaged using a ChemiDoc imaging system.

### 2.6. In Vitro Enzymatic Characterization of Purified rMIP_07528 and rPtpB-Mtb

Phosphatase activity of recombinant MIP_07528 and rPtpB-Mtb was quantitatively assessed using para-nitrophenyl phosphate (pNPP) as a substrate. Hydrolysis of pNPP by active phosphatases yields para-nitrophenol, a chromogenic product exhibiting maximum absorbance at 415 nm. The assay was performed in 96-well microplates, with each well containing a 150 μL reaction mixture. The reaction mixture consisted of 1.0 nmol of purified recombinant protein in reaction buffer (20 mM Tris, 150 mM NaCl, pH 7.5) with Milli-Q water. The reactions were initiated by adding pNPP to a final concentration of 20 Mm, and the reaction was incubated at 37 °C for 30 min. The absorbance at 415 nm was measured using an iMark™ Microplate Absorbance Reader (Bio-Rad). Subsequently, steady-state kinetic parameters (*K_m_* and *V_max_*) for *rMIP_07528* and *rPtpB-Mtb* were determined by measuring initial reaction velocities at varying substrate concentrations. The pNPP assay was performed with substrate concentrations ranging from 2 to 8 mM pNPP, ensuring coverage of the Michaelis-Menten curve. Initial velocities were calculated from the linear portions of reaction progress curves. The data were then fitted to the Michaelis-Menten equation using GraphPad Prism 9 software to determine *K_m_*, *V_max_*, *K_cat_*, and *K_cat_*/*K_m_* values. All measurements were performed in triplicate, and the result was expressed as mean values ± standard deviation.

### 2.7. Oxidative Inactivation Susceptibility Profile of Recombinant MIP_07528 Protein and rPtpB-Mtb

To evaluate differential oxidative inactivation between rMIP_07528 and rPtpB-Mtb, the recombinant proteins were immobilized on Ni-NTA resin to facilitate controlled reduction. Following immobilization, proteins were treated with 10 mM DTT in 25 mM Tris buffer (pH 7.5, 150 mM NaCl) for 20 min to ensure complete reduction in catalytic cysteine residues. Post-reduction, the resin was washed three times with DTT-free buffer to eliminate residual reductants, and proteins were eluted using 125 mM imidazole. Eluted proteins underwent dialysis against 10,000 volumes of fresh buffer to remove imidazole and trace DTT, ensuring no interference with subsequent oxidation steps.

For oxidative treatment, reduced enzymes were incubated with hydrogen peroxide (31.25–500 μM) for 20 min at room temperature. To assess pH-dependent activity, reaction mixtures were prepared at three distinct pH conditions of 5.5, 7.5 and 8.5. Phosphatase activity was quantified using 10 mM *p*-nitrophenylphosphate (pNPP) as a substrate. Reactions were quenched with 0.4 M NaOH, and hydrolysis rates were determined by measuring absorbance at 450 nm. Residual activity was normalized to untreated controls (100% activity) and expressed as mean ± standard deviation from triplicate experiments.

### 2.8. Evaluation of Immunomodulatory Effects of rMIP_07528 on THP-1 Cells

To assess the immunomodulatory potential of rMIP_07528, cytokine secretion profiles in THP-1 macrophage-like cells were evaluated using Enzyme-Linked Immunosorbent Assays (ELISA), in accordance with earlier reports [26,27]. THP-1 human monocytic cells were cultured in RPMI 1640 medium supplemented with 10% fetal bovine serum (FBS) and 1% penicillin/streptomycin. Cells were maintained at 37 °C in a humidified atmosphere with 5% CO2. To induce differentiation into macrophage-like cells, THP-1 cells were treated with 20 ng/mL phorbol 12-myristate 13-acetate (PMA) for 24 h [28,29]. Post-differentiation, cells (2 × 10^4^ cells/cm^2^) were treated with purified recombinant rMIP_07528 at concentrations of 2.5, 5, 10, and 20 μg/mL for 24 h. Control cells were either untreated or stimulated with lipopolysaccharide (LPS) as a positive control. Following treatment, cell culture supernatants were collected, and the levels of pro-inflammatory cytokines (TNF-α and IL-6) and the anti-inflammatory cytokine IL-10 were quantified using commercially available ELISA kits- Thermo Scientific, United states. Absorbance was measured using a BioTek Synergy HTX Multimode Reader- United States, at the appropriate wavelengths. Cytokine concentrations were determined from standard curves and expressed as mean ± standard deviation (SD) from three independent experiments [30].

## 3. Results

### 3.1. In Silico Analysis Identifies MIP_07528 as a Conserved Mycobacterial PtpB Ortholog

The sequence conservation of MIP_07528 among different *Mycobacterium* species was investigated through in silico analyses. Assessment of nucleotide and protein sequence identity between MIP_07528 and PtpB orthologs from *M. tuberculosis*, *M. avium*, *M. bovis*, and *M. smegmatis* revealed a high degree of conservation (Figure 1B). Specifically, nucleotide BLAST analysis indicated 77% identity between MIP_07528 and *ptpB*-Mtb, 77% identity between MIP_07528 and *ptpB*-*M.bovis*, 88% identity between MIP_07528 and *ptpB*-*M. avium*, and 69% identity between MIP_07528 and *ptpB*-*M. smegmatis*. Protein BLAST analysis demonstrated 72% identity between MIP_07528 and PtpB-Mtb, 72% identity between MIP_07528 and PtpB-*M. bovis* PtpB, 88% identity between MIP_07528 and PtpB-*M. avium*, and 55% identity between MIP_07528 and PtpB-*M. smegmatis* (Figure 1B).

Multiple sequence alignment further illustrated the conservation of key residues, particularly within the active site signature (CFAGKDRT), across these species (Figure 1A). Structural analysis of the predicted active sites revealed striking similarities in the three-dimensional architecture of catalytic pockets of MIP_07528 and all mycobacterial PtpBs examined. The catalytic cysteine and arginine residues, essential for phosphotyrosine substrate recognition and hydrolysis, displayed identical spatial orientations in MIP_07528 and the selected mycobacterial PtpB orthologs (Figure 1C).

Phylogenetic analysis, employing the neighbor-joining method with 1000 bootstrap replicates, demonstrated the evolutionary relationship of MIP_07528 with other mycobacterial PtpB orthologs (Figure 1D). The phylogenetic tree revealed that MIP_07528 is closely related to PtpB from the pathogenic *M. avium* and *M. leprae,* as well as *M. tuberculosis* and *M. bovis*. This close evolutionary relationship with pathogenic mycobacterial PtpBs suggests potential functional similarities, despite MIP’s non-pathogenic nature. The phylogenetic tree further revealed that mycobacterial PtpBs form a distinct clade separate from the outgroup *Nocardia farcinica*, reinforcing the specificity of this phosphatase family within the *Mycobacterium* genus.

The high sequence conservation, particularly within the active site, coupled with the phylogenetic relatedness to known PtpB orthologs, suggested that MIP_07528 is a functional PtpB ortholog in *M. indicus pranii*. This conservation implies that MIP_07528 likely possesses similar phosphotyrosine phosphatase activity and biological roles as its characterized counterparts in other mycobacterial species.

### 3.2. Cell Lysate and Culture Filtrate Has Phosphatase Activity in MIP

In silico analysis identified MIP_07528 as a putative protein tyrosine phosphatase (PtpB) in *M. indicus pranii* (MIP). To investigate its expression and enzymatic activity, phosphatase assays were performed on both cell lysates and culture filtrates of MIP. Measurable phosphatase activity was detected in both fractions, indicating the presence of an active phosphatase. To confirm the transcriptional expression of MIP_07528, qPCR was performed using gene-specific primers, verifying the presence of its mRNA. These findings established that MIP_07528 is actively transcribed in MIP, supporting its identification as an expressed phosphatase. Comparative phosphatase activity assays revealed differential phosphatase activity in MIP and *M. tuberculosis* (Mtb). In the culture filtrate, phosphatase activity in MIP was approximately 380 pmoles, significantly higher than the 220 pmoles observed in Mtb (Figure 2B(i)). Conversely, within the cell lysate, phosphatase activity in MIP was lower (470 pmoles) compared to Mtb (850 pmoles) (Figure 2B(ii)). This contrasting activity profile suggests it may be attributed to the variable growth rates of the two mycobacterial species. The comparatively rapid growth of MIP in culture may result in enhanced secretion levels of phosphatase as compared to the slow-growing Mtb. The higher levels of phosphatase activity observed in the cell lysate of Mtb as compared to MIP further confirm this hypothesis.

To assess the regulatory response of MIP_07528 under environmental stress, its expression was analyzed under oxidative (H_2_O_2_) and acidic (pH 5.5) conditions using real-time PCR. A dose-dependent increase in MIP_07528 expression was observed in response to increasing concentrations of H_2_O_2_, with transcript levels rising approximately 1.3-fold at 31.25 μM H_2_O_2_ and reaching 1.7-fold at 500 μM H_2_O_2_, relative to the control (Figure 2A(i)). In contrast, under acidic conditions (pH 5.5), MIP_07528 expression was slightly downregulated compared to neutral conditions (pH 7.5), with SigA serving as the internal control (Figure 2A(ii)).

These findings confirm that MIP_07528 is actively expressed as a phosphatase in MIP and its expression is modulated by environmental stressors.

### 3.3. Heterologous Expression and Purification of MIP_07528 and Rv0153c (ptpB-Mtb) in E. coli

The open reading frames of MIP_07528 and *Rv0153c* (*ptpB*-Mtb) were successfully cloned into the pET28a expression vector. Confirmation of successful cloning was achieved by restriction enzyme digestion analysis, which demonstrated the release of DNA fragments of approximately 831 bp corresponding to the sizes of MIP_07528 and *Rv0153c* (Figure 3A(i),B(i)). The recombinant pET28a constructs containing MIP_07528 or *Rv0153c* were transformed into *E. coli* BL21 (DE3) expression cells. The expression of recombinant proteins with a molecular weight of approximately 30 kDa, corresponding to the predicted size of His-tagged rMIP_07528 and rPtpB-Mtb, was observed in the induced (I) lanes (Figure 3A(ii),B(ii)). In contrast, no band was visible in the uninduced (U) lanes, indicating that IPTG induction effectively initiated the expression of the target proteins. The purity of the eluted proteins was confirmed by SDS-PAGE, revealing a single, prominent band at approximately 30 kDa in the purified (P) fraction (Figure 3A(iii),B(iii)).

Western blot analysis confirmed the presence of the His-tagged rMIP_07528 and rPtpB-Mtb in the induced cell pellet fractions and purified fractions (Figure 3C). A clear band was observed in the induced (I) and purified (P) lanes for both rMIP_07528 and rPtpB-Mtb, while no band was visible in the uninduced (U) lanes, further confirming the identity and purity of the recombinant proteins.

### 3.4. MIP_07528 Protein Displays Phosphatase Activity

The catalytic potential of the purified MIP_07528 protein was evaluated through spectrophotometric analysis employing para-nitrophenyl phosphate (pNPP) as a chromogenic substrate. Following incubation at physiological temperature (37 °C), reaction mixtures containing the recombinant enzyme exhibited a distinctive yellow chromophore development, which was notably absent in the enzyme-free control reactions. Quantitative assessment via absorbance measurements at λ = 405 nm revealed substantial phosphomonoesterase activity associated with MIP_07528 protein. These observations confirmed that the purified recombinant protein maintains structural integrity and exhibits significant phosphomonoesterase activity.

### 3.5. Kinetic Analysis Reveals Superior Catalytic Efficiency of rMIP_07528 Protein over rPtpB-Mtb

Comparative kinetic analysis of rMIP_07528 and rPtpB-Mtb was performed using pNPP as a substrate. Steady-state kinetics revealed that both enzymes exhibited Michaelis-Menten kinetics (Figure 4A). The kinetic parameters, including *V_max_*, *K_m_*, *K_cat_*, and *K_cat_*/*K_m_*, were determined for both enzymes (Figure 4B). The data demonstrated distinct differences in substrate affinity and catalytic efficiency between these mycobacterial phosphatases. For rPtpB-Mtb, the *V_max_* was 0.66 nmole/min/mg, the *K_m_* was 1.01 mM, the *K_cat_* was 0.036 s^−1^, and the *K_cat_*/*K_m_* was 0.36 M^−1^s^−1^. For rMIP_07528, the *V_max_* was 0.094 nmole/min/mg, the *K_m_* was 0.30 mM, the *Kcat* was 0.005 s^−1^, and the *K_cat_*/*K_m_* was 0.16 M^−1^s^−1^ (Figure 4B). rPtpB-Mtb exhibited an approximately 7-fold higher *V_max_* compared to rMIP_07528, while rMIP_0752 displayed a lower *K_m_*, suggesting a higher affinity for the substrate. The kinetic analysis reveals that both rMIP_07528 and rPtpB-Mtb are active phosphatases with distinct kinetic properties. However, the lower *K_cat_*/*K_m_* values of the recombinant mycobacterial proteins indicate that phosphatases in MIP have significantly lower catalytic activity as compared to those in Mtb.

### 3.6. Enhanced Susceptibility to Oxidative Inactivation of rMIP_07528 Compared to rPtpBMtb

The susceptibility of rMIP_07528 and rPtpB-Mtb to oxidative inactivation was assessed through controlled oxidation experiments. The structural differences between the two isoforms were highlighted by the presence of phenylalanine at position 222 in rPtpB-Mtb, which was absent in rMIP_07528, suggesting a potential role in their distinct oxidative resistance profiles (Figure 5A), as Phe222 has been previously reported to inhibit a The relative activities of rMIP_07528 and rPtpB-Mtb were measured at three different pH levels (5.5, 7.5, and 8.5) following exposure to varying concentrations of hydrogen peroxide (H_2_O_2_). As shown in Figure 5B(i), at pH 5.5, rMIP_07528 exhibited significantly reduced activity compared to rPtpB-Mtb after treatment with H_2_O_2_, indicating increased susceptibility to oxidative stress. At neutral pH (7.5), the trend persisted, with rMIP_07528 demonstrating lower residual phosphatase activity than rPtpB-Mtb (Figure 5B(ii)). The disparity became even more pronounced at pH 8.5, where rMIP_07528’s activity was markedly diminished in the presence of H_2_O_2_, further confirming its vulnerability to oxidative inactivation (Figure 5B(iii)).

Quantitative analysis revealed that the activity of rMIP_07528 decreased significantly with increasing concentrations of H_2_O_2_ across all tested pH levels, while rPtpB-Mtb maintained a higher level of activity under similar conditions. Specifically, rMIP_07528 demonstrated pronounced susceptibility to oxidation, with activity sharply decreasing to approximately 20% of control levels at the lowest tested H_2_O_2_ concentration (31.25 μM). This inhibitory effect intensified progressively with increasing H_2_O_2_ concentration, suggesting that rMIP_07528 possesses redox-sensitive catalytic or structural elements that undergo rapid oxidative modification even at relatively low peroxide concentrations. In contrast, rPtpB-Mtb exhibited considerably higher resistance to oxidative inactivation, retaining approximately 93% activity at 31.25 μM H_2_O_2_ and showing a more gradual reduction in activity at higher concentrations. Even at the highest tested peroxide concentration (500 μM), rPtpB-Mtb maintained substantial catalytic function, preserving approximately eight-fold higher activity than rMIP_07528 under identical conditions. These findings elucidate increased susceptibility of rMIP_07528 to oxidative inactivation compared to its rPtpB-Mtb, suggesting that structural differences may influence their functional resilience against oxidative inactivation.

### 3.7. rMIP_07528 Suppresses Pro-Inflammatory Cytokines in Macrophages

To investigate the potential immunomodulatory role of rMIP_07528, differentiated THP-1 cells were exposed to varying concentrations of rMIP_07528, and cytokine secretion profiles were analyzed. As depicted in Figure 6, treatment with rMIP_07528 induced significant alterations in the cytokine milieu. Exposure to rMIP_07528 resulted in a dose-dependent decrease in the secretion of pro-inflammatory cytokines TNF-α and IL-6 (Figure 6B,C). Statistical analysis revealed a significant reduction in TNF-α and IL-6 levels at higher doses of rMIP_07528 compared to untreated controls (*** *p* < 0.001). Concurrently, rMIP_07528 treatment led to a dose-dependent increase in the production of the anti-inflammatory cytokine IL-10 (Figure 6A), with statistically significant elevations observed at higher concentrations (*** *p* < 0.001).

These findings demonstrate that rMIP_07528 modulates cytokine secretion in THP-1 macrophage-like cells by suppressing the release of pro-inflammatory mediators and simultaneously enhancing the production of the anti-inflammatory cytokine IL-10. The observed immunomodulatory effects suggest a potential role for rMIP_07528in skewing the immune response towards an anti-inflammatory profile.

## 4. Discussion

Phosphatases have been reported to play key roles in the virulence of various microorganisms, influencing host cell signaling pathways and disrupting immune responses. These enzymes, often secreted by pathogenic bacteria, disrupt essential host processes, facilitating the survival and replication of pathogens within their hosts [23,31,32]. In particular, phosphoinositide (PI) metabolism is a critical target; pathogenic bacteria manipulate this pathway through phosphatases to evade immune defenses and enhance their pathogenicity [33]. Recent discoveries have identified a novel family of phosphatases characterized by a conserved P-loop signature, which exhibit phosphatase activity towards PIs and are distinct from human homologues, marking them as promising targets for therapeutic intervention. Specifically, in *Mycobacterium tuberculosis*, several phosphatases, such as PtpB-Mtb, play crucial roles in manipulating host cell environments to prevent phagosome maturation and promote bacterial survival [31]. In the hostile host microenvironment, elucidating these mechanisms provides crucial insights for developing anti-virulence strategies aimed at restoring native immune functions against persistent infections.

Investigating conserved phosphatases in species shifting from a saprophytic lifestyle to intracellular parasitism is essential for understanding the evolution of parasitism within mycobacterial lineages. The unique structural characteristics of these phosphatases may be linked to their unconventional biological functions, a relationship that warrants further investigation. Such explorations are particularly relevant for therapeutically significant microorganisms like MIP. Hence, to establish MIP_07528 as a homolog of PtpB, it was successfully cloned, expressed, purified, and characterized functionally through bioinformatics and in vitro analyses. The functional characterization of MIP_07528 from MIP as a tyrosine phosphatase contributes to this body of knowledge, particularly regarding its sensitivity to oxidative inactivation and its role in immunomodulation.

The functional characterization of MIP_07528 as a protein tyrosine phosphatase B (PtpB) homolog in *Mycobacterium indicus pranii* illuminates the evolutionary divergence of conserved virulence factors in non-pathogenic mycobacteria. Our in-depth bioinformatic analysis revealed striking sequence conservation patterns across the mycobacterial genus, with MIP_07528 sharing substantial nucleotide identities with pathogenic counterparts—77% with *M. tuberculosis* and *M. bovis*, 88% with *M. avium*—and a more distant yet significant 69% with the saprophytic *M. smegmatis*. These relationships manifest similarly at the protein level (72%, 88%, and 55% identity, respectively), suggesting evolutionary preservation of this phosphatase. The remarkable conservation of the catalytic architecture, particularly the signature active site motif (CFAGKDRT), was observed in our in silico analysis. These findings are consistent with previous investigations on PtpB-Mtb, in which this highly conserved motif was identified as essential for catalytic function and substrate specificity. The key residues within this motif—including Asp165, which functions as the general acid in the catalytic mechanism, and Lys164, which facilitates phosphoinositide binding—are meticulously preserved in MIP_07528, suggesting functional parallels with its pathogenic counterparts, particularly in substrate recognition and hydrolysis [31,34]. Prior investigations have highlighted intriguing structural parallels between the mycobacterial P-loop motif and those in eukaryotic 3-phosphatases like PTEN and myotubularin—despite their limited sequence homology—suggesting convergent evolution toward optimal phosphate hydrolysis architecture [35,36,37,38]. Our phylogenetic analysis positioned MIP_07528 within the clade of pathogenic mycobacterial phosphatases, distinctly separated from the *Nocardia farcinica* outgroup, reinforcing its ancestral relationship with virulence-associated phosphatases despite MIP’s non-pathogenic phenotype. This evolutionary conservation of the catalytically essential motif in MIP_07528, together with the subtle sequence variations detected in surrounding regions, likely explains its distinct substrate preferences and modified catalytic properties compared to its pathogenic counterparts— illustrating nature’s refinement of conserved enzymatic scaffolds to diversify biological functions while preserving fundamental catalytic efficiency.

Despite these structural similarities, our biochemical analyses uncovered fascinating functional divergences between the putative PtpB-MIP and its pathogenic ortholog. PtpB-MIP exhibited a contrasting compartmental activity profile —enhanced activity in culture filtrate (380 pmoles compared to PtpB-Mtb’s 220 pmoles) yet diminished activity within cell lysates (470 pmoles versus 850 pmoles for PtpB-Mtb). These observations may be correlated to the differential growth kinetics of the fast-growing MIP and slow-growing, virulent Mtb. The quantity of PtpB secreted into the culture would be presumably lower in the case of slow-growing pathogens like Mtb as compared to the fast-growing MIP. This hypothesis would also explain the reduced activity of PtpB-MIP in the cell lysate compared to that of Mtb [39].

Additionally, the expression of MIP_07528 exhibited a nuanced stress-responsive pattern—showing robust upregulation (1.7-fold increase) under oxidative challenge while demonstrating unexpected downregulation under acidic conditions (pH 5.5). This differential responsiveness hints at specialized adaptation to oxidative environments, potentially enhancing MIP’s persistence within the reactive oxygen species (ROS)-rich microenvironment of host macrophages. This differential expression profile parallels adaptive strategies observed in pathogenic mycobacterial phosphatases such as PtpB-Mtb, which modulates host phosphoinositide signaling during oxidative stress conditions. Studies have demonstrated that PtpB-Mtb’s interaction with phosphoinositides disrupts phagosomal maturation, potentially explaining the upregulation of structurally similar phosphatases under oxidative conditions [31].

Subtle yet significant alterations in catalytic behavior were revealed in the analysis of enzyme kinetic characterization of rMIP_07528 and rPtpB-Mtb. While both phosphatases adhere to classical Michaelis–Menten kinetics, rPtpB-MIP demonstrated markedly enhanced substrate affinity (Km = 0.30 mM) paired with substantially reduced catalytic efficiency (Vmax = 0.094 nmole/min/mg) compared to its *M. tuberculosis* counterpart (Km = 1.01 mM; Vmax = 0.66 nmole/min/mg). These kinetic parameters demonstrate a striking contrast to the versatile enzymatic profile of rPtpB-Mtb [19,34,40,41]. This substrate versatility allows pathogenic mycobacteria to simultaneously disrupt multiple host-signaling cascades during infection. The approximately two-fold reduction in catalytic turnover (Kcat/Km) observed in rPtpB-MIP not only correlates with MIP’s attenuated virulence profile but also suggests evolutionary refinement toward more specialized functions. Such alterations in kinetic properties frequently signal functional divergence, as documented in other bacterial phosphatases transitioning from virulence factors to immunomodulatory proteins. Our findings therefore suggest that these enzymatic modifications may have redirected PtpB-MIP’s functional role from aggressive pathogenesis toward nuanced immunomodulation, contributing to MIP’s distinctive host interaction profile.

Perhaps the most striking functional divergence emerged in our oxidative inactivation studies. rPtpB-MIP exhibited extraordinary vulnerability to H_2_O_2_ exposure across all examined pH conditions (5.5, 7.5, and 8.5), with this susceptibility becoming particularly pronounced in alkaline environments. When exposed to even minimal hydrogen peroxide concentrations (31.25 μM H_2_O_2_), rPtpB-MIP activity plummeted to merely 20% of baseline levels, while rPtpB-Mtb maintained a remarkable 93% of its catalytic function under identical conditions. This dramatic sensitivity escalated with increasing oxidant concentrations, culminating in an approximately eight-fold difference in residual activity between the enzymes at 500 μM H_2_O_2_. These findings align with previous studies where PtpB-Mtb exhibited notable resistance to oxidative inactivation [32,42]. This resistance is believed to be a molecular adaptation in pathogenic mycobacteria to survive the oxidative burst mounted by the host’s immune system. The oxidative microenvironment within host macrophages, characterized by reactive oxygen species (ROS) generation during respiratory bursts, represents a critical defense mechanism against intracellular pathogens like *Mycobacterium tuberculosis* [43,44]. Mechanistically, the heightened susceptibility of MIP_07528 can be attributed to the catalytic cysteine, which is highly nucleophilic and particularly prone to oxidation by host-derived ROS and RNS. Oxidative inactivation of PTPs commonly proceeds through the formation of sulfenic acid at the catalytic cysteine, which may further oxidize to sulfinic or sulfonic acid or lead to disulfide bond formation and oligomerization, resulting in reversible or irreversible loss of enzymatic activity. Mycobacterial PtpB has been earlier reported to circumvent this limitation through dynamic structural adaptations, particularly via its unique lid domain that confers oxidative resistance [45,46,47,48]. This evolutionary innovation, observed exclusively in mycobacterial PtpB homologs, suggests specialized optimization to withstand host-derived oxidative stress while maintaining phosphatase activity. A kinetic model suggested that only the open conformation of PtpB is susceptible to inactivation by H_2_O_2_. The rapid conformational switching of the lid limits the access of ROS to the active site, thus preserving enzyme activity. Interestingly, in silico analysis of MIP_07528 revealed the absence of a specific phenylalanine residue at position 222, suggesting conformational changes in the two-helix lid structure in rPtpB-MIP that shields the active site from ROS-mediated damage in Mtb. Previous studies have reported Phe222 to serve as the molecular gatekeeper regulating substrate access and protecting the catalytic cysteine in mycobacterial PtpB from oxidative damage [32,42]. The absence of this protective feature in MIP_07528 likely represents a regulatory adaptation, rendering its activity sensitive to the redox state of the microenvironment and restricting its function to conditions of low oxidative stress. This redox-sensitive regulation may underpin a transient immunomodulatory role, consistent with the non-pathogenic and therapeutic profile of MIP, in contrast to the persistent immune evasion strategies employed by pathogenic mycobacteria.

Previous studies have reported enhanced sensitivity to oxidative inactivation in lidless PTPs like PtpA and YopH, reinforcing the protective role of the PtpB-Mtb lid [32,49,50]. The kinetic characterization of rPtpB-MIP further highlights its functional divergence from pathogenic PtpB orthologs. rPtpB-MIP displays higher substrate affinity (Km = 2.4 μM) but is markedly more susceptible to oxidative inactivation than rPtpB-Mtb (Km = 5.8 μM). These combined kinetic and structural features suggest that rPtpB-MIP is adapted for rapid and dynamic regulation of phosphatase activity that is sensitive to the oxidative state of the host microenvironment. Such adaptability may underpin its role in transient immunomodulation, supporting beneficial host–microbe interactions characteristic of non-pathogenic MIP, rather than the persistent immune evasion strategies employed by pathogenic mycobacteria [34,39,51]. This evolutionary divergence in enzymatic properties underscores the specialization of PtpB homologs for distinct roles in host–pathogen versus host–commensal interactions. The immunomodulatory capabilities of rPtpB-MIP provide compelling evidence for its functional divergence. In experiments with differentiated THP-1 macrophage-like cells, rPtpB-MIP’s remarkable capacity to orchestrate a comprehensive shift in the cytokine milieu was revealed, transitioning from pro-inflammatory to anti-inflammatory profiles. Dose-dependent suppression of the pro-inflammatory mediators TNF-α and IL-6 was induced upon exposure to increasing concentrations of rPtpB-MIP, with statistically significant reductions (*p* < 0.001) observed at higher treatment doses. This inflammatory dampening was found to occur concomitantly with significant enhancement of IL-10 secretion, a critical anti-inflammatory cytokine. The specificity and physiological relevance of these immunomodulatory effects were validated by the robust responsiveness of the THP-1 cell model to lipopolysaccharide challenge, as evidenced by appropriate elevations in pro-inflammatory cytokines. Intriguing parallels can be drawn from these observations by comparing them with the well-characterized immunomodulatory functions of PtpB-Mtb, suggesting evolutionary conservation alongside functional divergence. PtpB-Mtb is known to serve as a critical virulence factor in *M. tuberculosis* that subverts host immunity by suppressing pro-inflammatory cytokines, including IL-6 and IL-1β, through disruption of TLR/MyD88 and MAPK signaling pathways, without affecting JAK2, STAT1, or JNK pathways [19,20,22,45,51,52,53]. A similar potent suppression of TNF-α and IL-6 was exhibited by rPtpB-MIP, suggesting conservation of this fundamental immunomodulatory capacity. While both rPtpB-MIP and PtpB-Mtb suppress TNF-α and IL-6, the context and likely biological outcomes differ. PtpB-Mtb primarily functions to evade immune clearance and promote pathogen survival [54,55]; it is suggested by the present data that rPtpB-MIP may have been playing a key role in MIP to establish beneficial host–microbe interactions, consistent with the non-pathogenic and therapeutic profile of MIP. Notably, the pronounced upregulation of IL-10 by PtpB-MIP points toward a mechanism favoring immune regulation and non-pathogenic coexistence rather than antagonistic pathogenesis. The cytokine modulation pattern observed with rPtpB-MIP suggests engagement of signaling pathways similar to those targeted by PtpB-Mtb, such as MAPK and NF-κB, but potentially yielding distinct functional outcomes [46,48]. This is further supported by the evolutionary conservation of catalytic domains between MIP_07528 and pathogenic mycobacterial PtpB orthologs. However, direct evidence for the involvement of specific pathways, such as MAPK, NF-κB, or STATs, in MIP_07528-mediated immunomodulation is currently lacking. Across bacterial species, phosphorylation systems shape regulatory networks, as seen in *Streptococcus pyogenes*, where a non-typical tyrosine kinase affects virulence by phosphorylating CovR and other cellular targets [56]. PtpB-MIP’s dual capacity to reduce pro-inflammatory cytokines while boosting IL-10 points towards selective modulation of specific signaling branches, creating an anti-inflammatory milieu without compromising broader immune competence. Further studies employing pathway-specific inhibitors, reporter assays, and phosphoproteomic analyses will be essential to delineate the precise molecular targets and signaling mechanisms underlying these effects.

From an evolutionary standpoint, PtpB-MIP exemplifies functional divergence of tyrosine phosphatases, transitioning from virulence factors in pathogenic mycobacteria to immunomodulators in MIP. Unlike PtpB-Mtb, which facilitates immune evasion, PtpB-MIP promotes host–microbe equilibrium, potentially underpinning MIP’s non-pathogenicity despite shared immune-modulating homologs [57]. This divergence is further supported by genomic adaptations, including toxin–antitoxin (TA) modules that enhance regulatory flexibility, as observed in broader genomic studies of MIP [58]. Therapeutically, PtpB-MIP demonstrates dual anti-inflammatory and immunoregulatory properties, positioning it as a promising candidate for managing inflammatory and autoimmune disorders, contrasting with strategies targeting PtpB-Mtb inhibition to restore antimicrobial defenses [23].

In summary, our comprehensive characterization of MIP_07528 in *Mycobacterium indicus pranii* confirms its identity as a functional protein tyrosine phosphatase (PtpB) ortholog. Through in silico analyses, we have highlighted its conserved active site and phylogenetic relationship to various mycobacterial PtpBs, hinting at shared catalytic mechanisms. Biochemical assays validate its phosphatase activity, modulated by environmental stressors like oxidative and acidic conditions, further supported by gene expression studies. Comparative enzymatic kinetics revealed distinct activity profiles, with recombinant MIP_07528 exhibiting heightened susceptibility to oxidative inactivation compared to rPtpB-Mtb. Importantly, we demonstrate that rMIP_07528 exerts immunomodulatory effects on macrophages by suppressing pro-inflammatory cytokines and promoting IL-10 production. This study enriches our understanding of MIP_07528’s role in mycobacterial physiology and lays the groundwork for exploring its potential as a target for novel immunomodulatory strategies. Further structural studies, combined with in vivo infection models, will be crucial to fully elucidate the physiological relevance of PtpB-MIP’s immunomodulatory activity.

The characterization of MIP_07528 as an immunomodulatory phosphatase in *Mycobacterium indicus pranii* opens several avenues for therapeutic exploration. Given MIP’s established immunotherapeutic effects, further investigation into the specific substrates and regulatory mechanisms of MIP_07528 may reveal novel nodes of immune modulation. Manipulating MIP_07528 activity could enable fine-tuning of macrophage responses—potentially enhancing beneficial immune pathways or mitigating excessive inflammation—depending on its role within the broader immunomodulatory network of MIP. If MIP_07528 exhibits substrate preferences or regulatory features distinct from pathogenic orthologs, it may serve as a unique tool for modulating macrophage function without cytotoxicity, supporting its application as an adjuvant in vaccines or as a modulator in autoimmune and inflammatory diseases [19,50,59,60].

Identification of MIP_07528’s host targets, including both protein and lipid substrates, could uncover new intervention points for therapeutic manipulation. These insights may inform the rational design of small-molecule modulators that selectively target bacterial phosphatases, minimizing off-target effects on host enzymes. Notably, the increased redox sensitivity and absence of a protective lid domain in MIP_07528, in contrast to the oxidative resilience of pathogenic mycobacterial PtpBs, provide a structural framework for the development of selective inhibitors or activators.

Given the conservation of PtpB orthologs in various human pathogens, structural and functional studies of MIP_07528 may facilitate the design of broad-spectrum antivirulence agents. Selective targeting of pathogen-specific phosphatase features could help restore host immunity in infectious diseases. Furthermore, as bacterial protein tyrosine phosphatases have been implicated in the regulation of macrophage phenotypes relevant to cancer and autoimmune disorders, further characterization of MIP_07528 may inform strategies for modulating the tumor microenvironment or correcting immune dysregulation [19,45,50,51,61]. Future research should focus on in vivo validation, structural elucidation of substrate interactions, and the translational assessment of MIP_07528-targeted interventions for infectious, neoplastic, and autoimmune diseases.

## 5. Conclusions

In the present study, MIP_07528 was characterized as a functional protein tyrosine phosphatase B (PtpB) ortholog in MIP through comprehensive bioinformatic, biochemical, and immunological approaches. Significant sequence conservation with pathogenic mycobacterial PtpB proteins was observed, particularly within the catalytic domain containing the signature P-loop motif. Despite these structural similarities, functional studies revealed distinctive properties of MIP_07528 compared to its *M. tuberculosis* counterpart, including differential compartmental activity profiles, stress responsiveness, and enzymatic kinetics. The enhanced susceptibility of MIP_07528 to oxidative inactivation represents a striking functional divergence from PtpB-Mtb, potentially attributable to structural variations in the protective lid domain. This evolutionary adaptation may reflect MIP’s non-pathogenic lifestyle, where survival within hostile oxidative environments is less critical than for pathogenic mycobacteria. The immunomodulatory activity of recombinant MIP_07528 was demonstrated through its ability to suppress pro-inflammatory cytokines while enhancing anti-inflammatory IL-10 production in macrophages. A comprehensive graphical summary of the study’s main findings has been illustrated in Figure 7.

These findings illuminate the evolutionary trajectory of PtpB from a virulence factor in pathogenic mycobacteria to an immunomodulator in non-pathogenic MIP. The functional characterization of MIP_07528 enhances our understanding of phosphatase-mediated immunomodulation in mycobacteria and provides insights into the molecular basis of MIP’s therapeutic potential. Future structural studies and in vivo infection models will be valuable to fully elucidate the physiological relevance of PtpB-MIP’s immunomodulatory activity and to explore its potential applications in managing inflammatory and autoimmune disorders.

## Figures and Tables

**Figure 1 biology-14-00565-f001:**
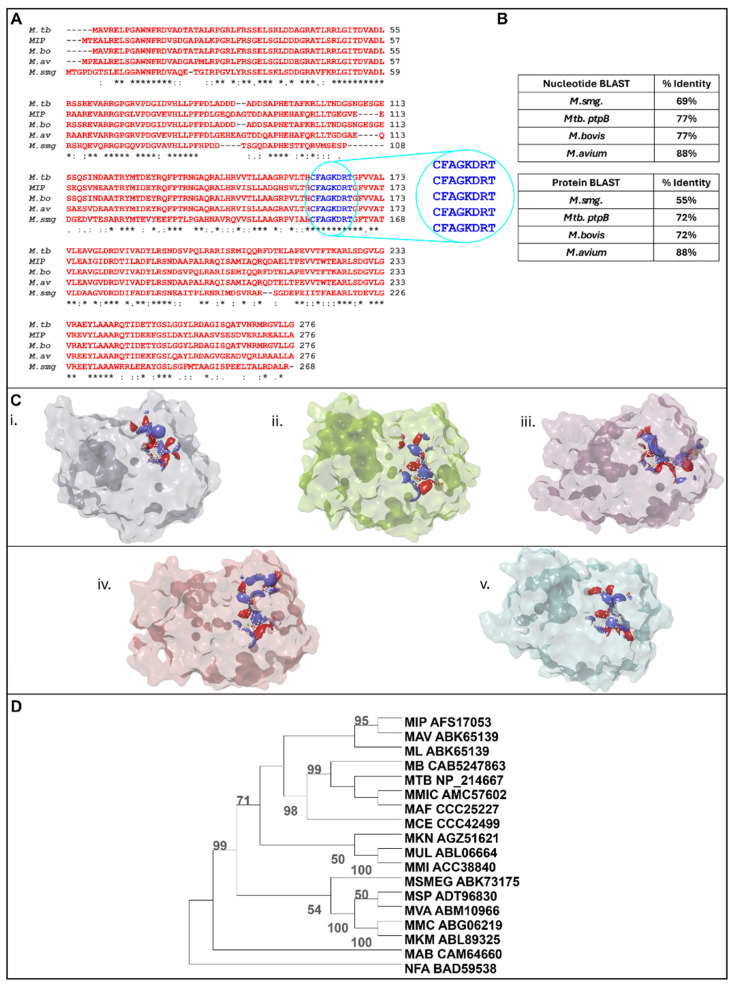
The sequence of MIPPtpB (*MIP_07528*) is conserved among different mycobacterium species. (**A**) Alignment of protein tyrosine phosphatase B sequences of *Mycobacterium tuberculosis* H37Rv (NP_214667), *Mycobacterium avium* (WP_009979776), *Mycobacterium bovis* (CAB5247863) and *Mycobacterium smegmatis* (ABK73175) with *Mycobacterium indicus pranii* (AFS17053); the encircled region shows the active site signature. (* shows conserved base pairs and circle shows the variable region in sequence allignment) (**B**) Nucleotide and protein BLAST study among the identified protein tyrosine phosphatase B of *Mtb*, *M. bovis*, *M. avium*, *M. smeg* and the putative *MIP_07528* to check the percentage identity among the pathogenic and the non-pathogenic mycobacterial species. (**C**) Active sites determined for (**i**) *Rv0153c,* (**ii**) *MIP_07528*, (**iii**) *Mb0158c,* (**iv**) *MAV_5145* and (**v**) *Msmeg_0100.* (**D**) Phylogenetic tree based on the maximum likelihood statistical method using the bootstrap method phylogeny test, showing conservation of PtpB across the genus of Mycobacterium. The nodes that were supported by bootstrap values > 50% (1000 replicates) are shown. *Nocardia farcinica* (NFA) was used as the outgroup. Abbreviations—MIP, *Mycobacterium indicus pranii*; MAV, *Mycobacterium avium complex*; ML, *Mycobacterium leprae*; *MB, Mycobacterium bovis*; MTB, *Mycobacterium tuberculosis H37Rv*; MMIC, *Mycobacterium tuberculosis variant microti*; MAF, *M. africanum*; MCE, *Mycobacterium canetti*; MKN, *Mycobacterium kansasii*; MUL, *Mycobacterium ulcerans*; MMI, *Mycobacterium marinum M*; MSMEG, *Mycobacterium smegmatis*; MSP, *Mycobacterium gilvum Spyr*; MVA, Mycobacterium vanbaalenii PYR-1; MMC, *Mycobacterium* sp. MCS; MKM, *Mycobacterium* sp. KMS; MAB, *Mycobacterium abscessus*.

**Figure 2 biology-14-00565-f002:**
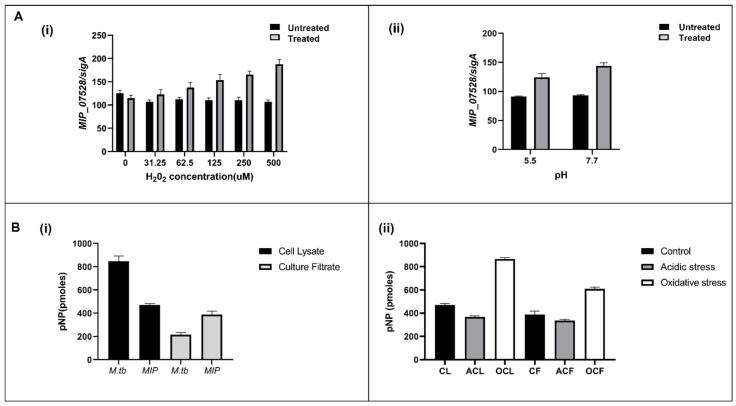
Expression and activity profiling of MIP_07528 under stress conditions. (**A**) Relative expression analysis of *MIP_07528* normalized to *sigA* under different stress conditions in *MIP* by qRT-PCR. (**i**) Expression under increasing concentrations of H_2_O_2_ (oxidative stress) shows a dose-dependent upregulation in treated samples compared to untreated controls. (**ii**) Expression under acidic (pH 5.5) and neutral-control (pH 7.7) conditions shows enhanced expression at alkaline pH. (**B**) *p*-Nitrophenyl phosphate (pNPP) release assay indicating phosphatase activity under various conditions. (**i**) Comparison of phosphatase activity in cell lysate and culture filtrate of *M.tb* and *MIP*. *M.tb* cell lysate shows the highest activity, followed by *MIP* lysate and culture filtrates. (**ii**) Phosphatase activity of *MIP* cell lysates (CL) and culture filtrates (CF) under control (C), acidic stress (AC), and oxidative stress (OC) conditions. Oxidative stress significantly increases activity in culture filtrate, suggesting stress-induced secretion or activation. Error bars represent standard deviation from three independent experiments.

**Figure 3 biology-14-00565-f003:**
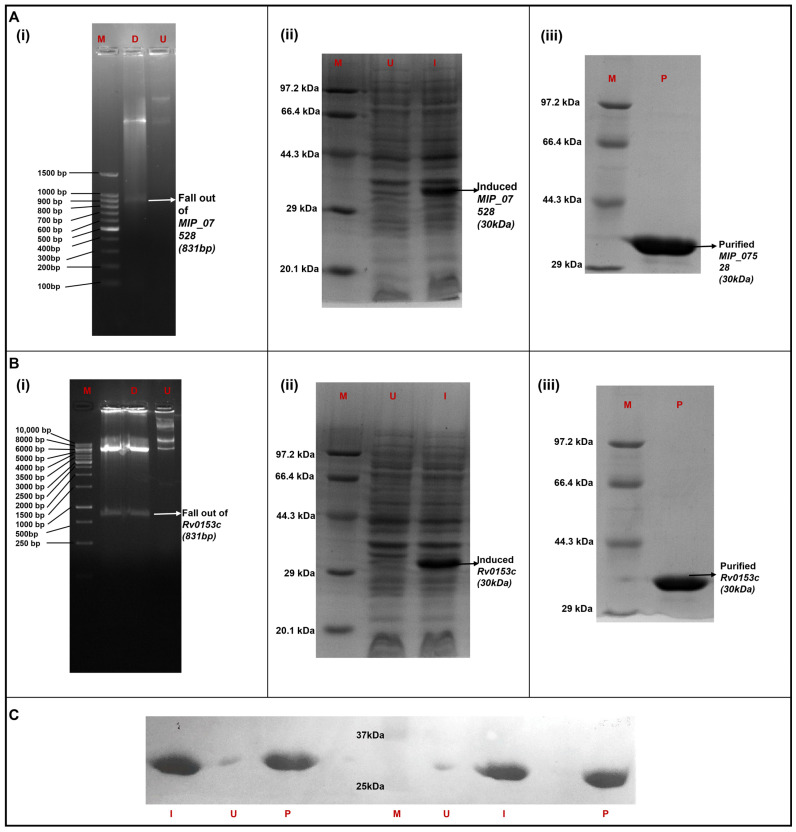
Cloning, expression, purification and Western blot for rMIPPtpB (**A**). (**B**) (**i**) Confirmation of cloning by restriction digestion of the recombinant plasmid containing MIP_07528 and Rv0153c, respectively, ligated in pET28a vector with BamHI and HindIII enzymes. (M) Marker (D) digested plasmid displaying a fallout at the desired gene size of 831 bp of MIP_07528 and Rv0153c, respectively. (U) Undigested recombinant plasmid containing MIP_07528 and Rv0153c, respectively, on a 1% agarose gel (**A**). (**B**) (**ii**) rMIPPtpB and rMTBPtpB protein expression, respectively, via IPTG induction. (M) Marker (U) uninduced culture pellet with no IPTG. (I) Induced culture pellet with 0.1 mM IPTG induction for 20 h at 20 °C, showing the expression of the desired protein (**A**). (**B**) (**iii**) Purification of both rMIPPtpB and rMTBPtpB, respectively, was carried out using Ni-NTA affinity chromatography. (M) marker (P) purified fraction of rMIPPtpB protein. (**C**) Western blot of His-tagged rMIPPtpB and rMTBPtpB left to right: (I) induced cell pellet fraction of rMTBPtpB, (U) uninduced cell pellet fraction of rMTBPtpB, (P) purified protein rMTBPtpB, (M) marker, (U) uninduced cell pellet fraction of rMIPPtpB, (I) induced cell pellet fraction of rMIPPtpB and (P) purified protein rMIPPtpB. The uncropped western blot figures were presented in Appendix A.

**Figure 4 biology-14-00565-f004:**
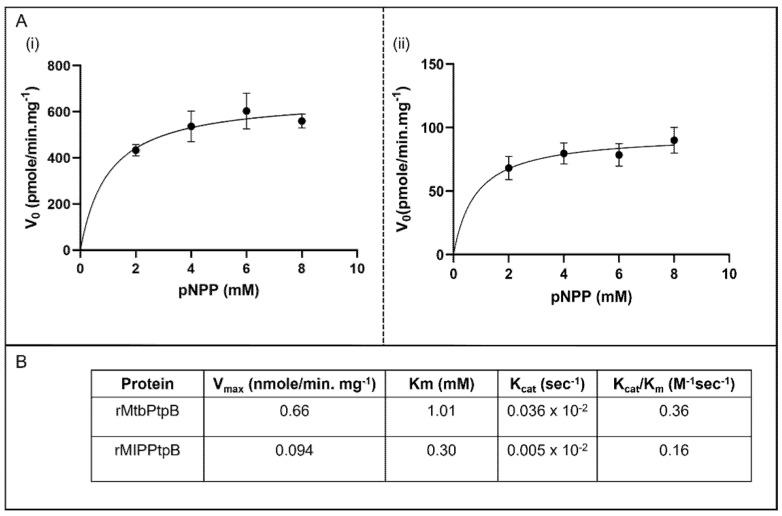
Comparative analysis of mycobacterial protein tyrosine phosphatases. (**A**) (**i**) Steady-state kinetics of rMTBPtpA and (**ii**) steady-state kinetics of rMIPPtpB were determined using the pNPP assay. The rate of substrate (pNPP) hydrolysis is plotted against increasing substrate concentrations. Data represent the mean ± SD of three independent experiments. (**B**) Kinetic analysis of mycobacterial protein tyrosine phosphatases.

**Figure 5 biology-14-00565-f005:**
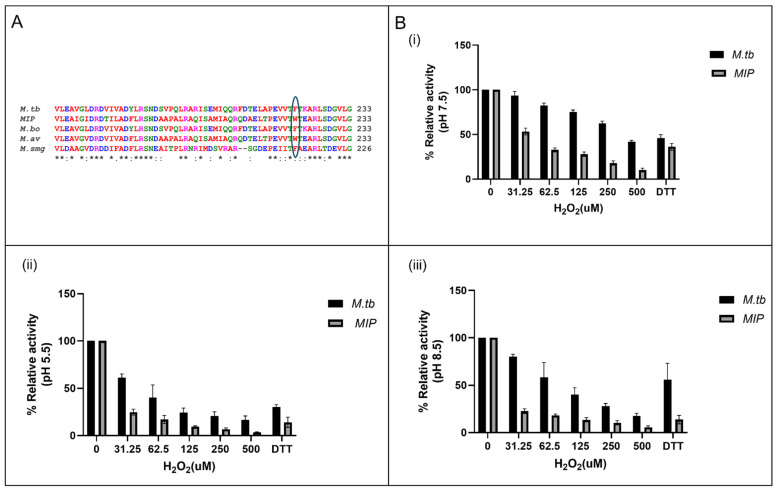
rPtpB (*M.tb*) resists H_2_O_2_ oxidation, but rPtpB (*MIP*) is more susceptible to oxidation by H_2_O_2_. (**A**) Phenylalanine is present at 222 in rPtpB (*M.tb*) but absent in rPtpB (*MIP*) at the same position. (* shows conserved base pairs and circle shows the variable region in sequence allignment) (**B**) (**i**) Relative activities of rPtpB (*MIP*) and rPtpB (*M.tb*) at pH 5.5. (**ii**) Relative activities of rPtpB (*MIP*) and rPtpB (*M.tb*) at pH 7.5. (**iii**) Relative activities of rPtpB (*MIP*) and rPtpB (*M.tb*) at pH 8.5. rPtpB (*MIP*) and rPtpB (*M.tb*) activities were assessed by monitoring the photometric detection of pNPP hydrolysis following a 20 min exposure to varying concentrations of H_2_O_2_. Error bars represent the standard deviation of triplicate measurements. rPtpB (*MIP*) exhibited susceptibility to oxidation compared to rPtpB (*M.tb*).

**Figure 6 biology-14-00565-f006:**
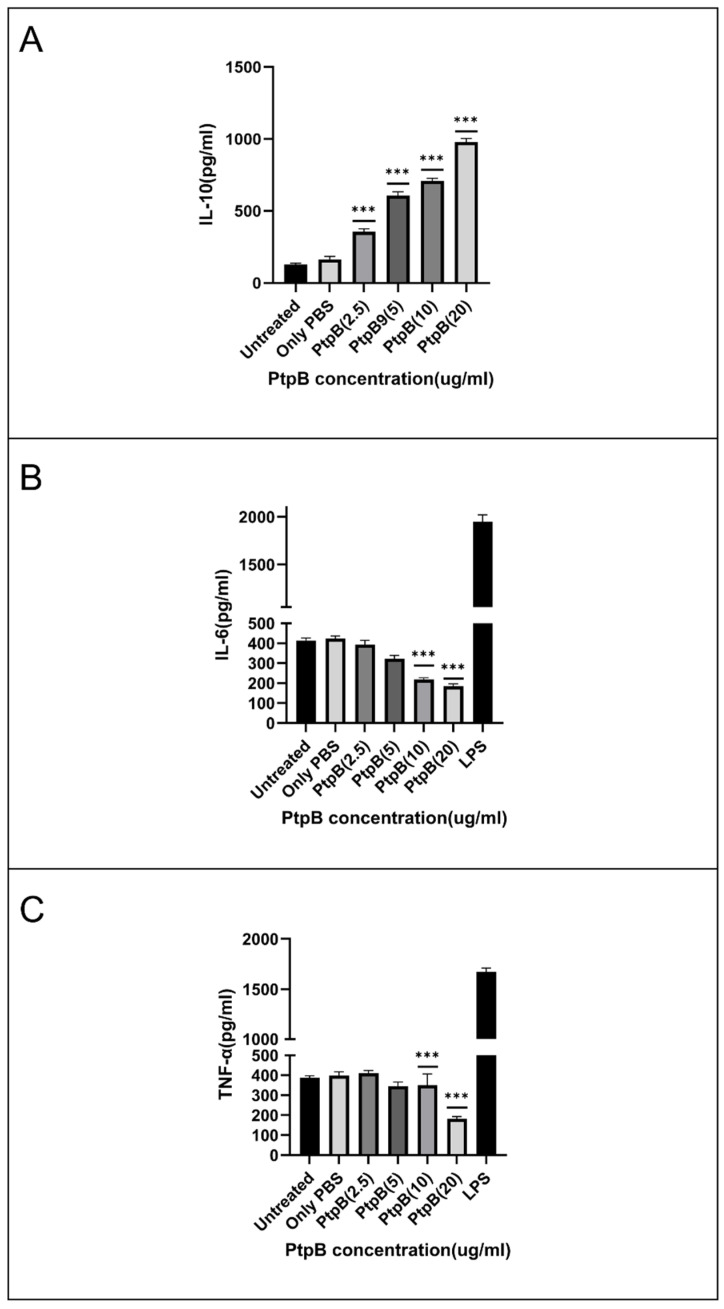
**Dose-dependent immunomodulatory effect of rPtpB (MIP) on cytokine secretion in THP-1 macrophage-like cells-** The rPtpB (*MIP*) modulates cytokine secretion in THP-1 macrophage-like cells by significantly reducing the levels of pro-inflammatory cytokines TNF-α and IL-6, while simultaneously increasing the release of the anti-inflammatory cytokine IL-10. (**A**) **IL-10 secretion**: rPtpB (MIP) treatment significantly increased the secretion of anti-inflammatory cytokine IL-10 in a dose-dependent manner compared to the untreated control. (**B**) **IL-6 secretion**: A significant dose-dependent reduction in the pro-inflammatory cytokine IL-6 was observed following treatment with increasing concentrations of rPtpB (MIP). (**C**) **TNF-α secretion**: Treatment with rPtpB (MIP) led to a marked decrease in TNF-α secretion, again showing a dose-dependent trend. This effect was observed after 24 h of treatment with various concentrations of rPtpB (*MIP*) (2.5, 5, 10, and 20 μg). The data, expressed as the mean ± SD from three independent experiments, showed a statistically significant reduction in TNF-α and IL-6 secretion (*** *p* < 0.001). Lipopolysaccharide (LPS) served as a positive control, yielding mean TNF-α and IL-6 levels of 1750 ± 50 pg/mL and 1943 ± 272 pg/mL, respectively. These findings suggest a dose-dependent immunomodulatory role of rPtpB in skewing the immune response towards an anti-inflammatory profile.

**Figure 7 biology-14-00565-f007:**
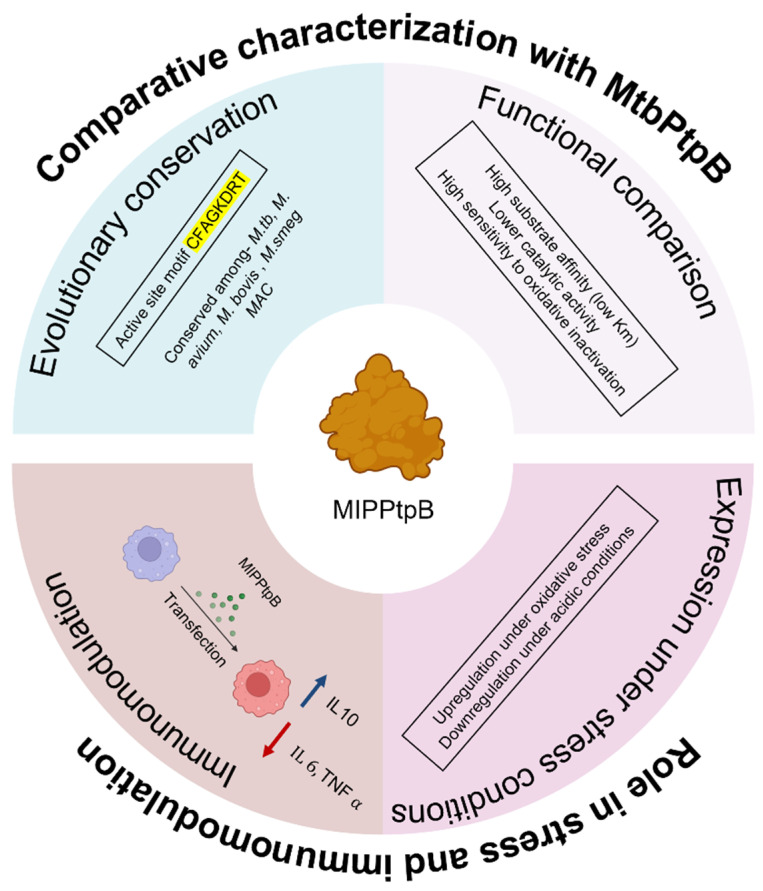
**Functional summary of MIP_07528 from *MIP*-Comparative characterization with MtbPtpB**—MIP_07528 maintains high sequence identity with PtpB-Mtb and conserved catalytic motif (CFAGKDRT). It shows higher substrate affinity but lower catalytic efficiency and dramatically reduced oxidative resistance compared to PtpB-Mtb. Role in stress and immunomodulation—MIPPtpB shows upregulation under oxidative stress with an enhanced secretion profile. It demonstrates immunomodulatory activity by suppressing pro-inflammatory cytokines (TNF-α, IL-6) while enhancing anti-inflammatory IL-10.

**Table 1 biology-14-00565-t001:** List of oligonucleotides used in this study.

Sr No.	Primer	Sequence (5′-3′)
1	MIP_07528 FP	AAAGGATCCATGACTGAGGCGTTGCGA
2	MIP_07528 RP	AAAAAGCTTTCAGGCGAGCAGCGCCTC
3	Rv0153c FP	AAAGGATCCATGGCTGTCCGTGAACTGC
4	Rv0153 RP	AAAAAGCTTTCATCCGAGCAGCACCCCGCG

## Data Availability

The original contributions presented in this study are included in the article. Further inquiries can be directed to the corresponding author(s).

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
