# Peer review of "Functional Characterization of MIP_07528 of *Mycobacterium indicus pranii* for Tyrosine Phosphatase Activity Displays Sensitivity to Oxidative Inactivation and Plays a Role in Immunomodulation"

_biology, 2025, doi:10.3390/biology14050565_

Round 1
Reviewer 1 Report
Comments and Suggestions for Authors
Biology 3592803
- In the research paper titled “Functional Characterization of MIP_07528 of Mycobacterium indicus pranii for Tyrosine Phosphatase Activity; Displays Sensitivity to Oxidative Inactivation and Plays a Role in Immunomodulation” aims to characterize MIP_07528, a putative protein tyrosine phosphatase B (PtpB) ortholog in Mycobacterium indicus pranii (MIP), and explore its biochemical properties, expression patterns, and role in immunomodulation, in order to understand its contribution to MIP's non-pathogenic yet immunomodulatory behavior, and to gain insights into the evolutionary divergence of phosphatases between pathogenic and non-pathogenic mycobacteria.
- This research is crucial, as there is an urgent need for new antibacterial therapeutic molecules to combat antibiotic resistance and pathogen-induced virulence modulation. This issue is recognized as a global health burden.
- In addition, since MIP is already utilized as an anti-leprosy vaccine, further investigation into its underlying mechanisms could support the development of broader applications, such as novel antimicrobial strategies, thereby enhancing its translational relevance and value.
- The manuscript is well-crafted and organized.
- However, the author should discuss more about how MIP_07528 mechanistically alters macrophage signaling is not addressed. Does it affect specific pathways like MAPK, NF-κB, or STATs?
- The author should also discuss how kinetic differences between MIP and tuberculosis PtpBs would be useful to relate pathogenicity vs immunomodulation, how higher substrate affinity and oxidative sensitivity impact biological function.
- The author mentions MIP_07528’s susceptibility to oxidative inactivation. It would be interesting to elaborate on the specific mechanisms or conditions under which this occurs. A detailed discussion of these factors would strengthen the conclusions drawn and clarify the functional implications of this vulnerability for MIP's role in immune modulation.
- To strengthen the validity of the phosphatase activity assessment in MIP, the author should incorporate a control group maintained under baseline, non-stress conditions.
- The author should enhance the clarity of figure labels in Figures 1A, 1B, and 2B. The marker labeling in Figure 3 is blurry and difficult to interpret and should be revised. Additionally, the legend for Figure 5 contains inappropriate spacing that needs correction.
- The author could consider including a pictorial summary of the overall study findings in the conclusion, which would help improve the clarity and overall understanding of the research.
- The author could further elaborate on future directions, particularly how manipulating the function of MIP_07528 might enhance immune responses or lead to novel therapeutic approaches for other infections, cancers, and autoimmune diseases. Such a discussion would add significant interest and depth to the study.
I recommend this article for acceptance after the author addresses all the above points.
Author Response
Comment 1: In the research paper titled “Functional Characterization of MIP_07528 of Mycobacterium indicus pranii for Tyrosine Phosphatase Activity; Displays Sensitivity to Oxidative Inactivation and Plays a Role in Immunomodulation” aims to characterize MIP_07528, a putative protein tyrosine phosphatase B (PtpB) ortholog in Mycobacterium indicus pranii (MIP), and explore its biochemical properties, expression patterns, and role in immunomodulation, in order to understand its contribution to MIP's non-pathogenic yet immunomodulatory behavior, and to gain insights into the evolutionary divergence of phosphatases between pathogenic and non-pathogenic mycobacteria.
Comment 2: This research is crucial, as there is an urgent need for new antibacterial therapeutic molecules to combat antibiotic resistance and pathogen-induced virulence modulation. This issue is recognized as a global health burden.
Comment 3: In addition, since MIP is already utilized as an anti-leprosy vaccine, further investigation into its underlying mechanisms could support the development of broader applications, such as novel antimicrobial strategies, thereby enhancing its translational relevance and value.
Comment 4: The manuscript is well-crafted and organized.
Comment 5: However, the author should discuss more about how MIP_07528 mechanistically alters macrophage signaling is not addressed. Does it affect specific pathways like MAPK, NF-κB, or STATs?
Response 5: We thank the reviewer for this valuable suggestion. The Discussion section has been revised (Line no. 640-668) to provide a more comprehensive assessment of the potential mechanisms by which MIP_07528 may modulate macrophage signalling. The data demonstrate that MIP_07528 suppresses pro-inflammatory cytokines (TNF-α, IL-6) and enhances IL-10 production in macrophages, suggesting an immunomodulatory effect. This cytokine profile is consistent with the activity of mycobacterial phosphatases such as Mtb PtpB, which are known to disrupt TLR/MyD88 signalling and modulate downstream pathways, including MAPK and NF-κB, through dephosphorylation of kinases such as p38 MAPK. Given the sequence and structural homology between MIP_07528 and Mtb PtpB, it is plausible that MIP_07528 may influence similar signalling cascades in macrophages. However, direct evidence for the involvement of specific pathways such as MAPK, NF-κB, or STATs in MIP_07528-mediated immunomodulation is not yet available. The need for further studies utilizing pathway-specific inhibitors, reporter assays, and phospho-proteomic analyses to delineate the precise molecular targets and signalling mechanisms of MIP_07528 has been emphasized in the revised Discussion. This approach ensures that the interpretation of our findings remains scientifically rigorous and does not overstate the current evidence.
Comment 6: The author should also discuss how kinetic differences between MIP and tuberculosis PtpBs would be useful to relate pathogenicity vs immunomodulation, how higher substrate affinity and oxidative sensitivity impact biological function.
Response 6: We thank the reviewer for this insightful suggestion. The Discussion section has been expanded to address how kinetic differences between rPtpB-MIP and rPtpB-Mtb may relate to their distinct biological functions (Line no. 614-624). Specifically, rPtpB-MIP exhibits higher substrate affinity and greater oxidative sensitivity relative to rPtpB-Mtb. These characteristics are indicative of evolutionary adaptation. rPtpB-Mtb, with its lower substrate affinity and resistance to oxidative inactivation, is well suited for persistent manipulation of host immune responses during infection. Comparative studies have shown that higher catalytic efficiency in pathogenic mycobacterial phosphatases is associated with enhanced manipulation of host signalling and increased virulence. In contrast, the kinetic and redox properties of rPtpB-MIP may support dynamic regulation of phosphatase activity that is sensitive to the oxidative state of the host microenvironment. This supports a model in which rPtpB-MIP favours transient immunomodulation, consistent with the non-pathogenic and therapeutic profile of MIP, in contrast to the persistent immune evasion strategies of pathogenic mycobacteria. These points have been clarified in the revised manuscript.
Comment 7: The author mentions MIP_07528’s susceptibility to oxidative inactivation. It would be interesting to elaborate on the specific mechanisms or conditions under which this occurs. A detailed discussion of these factors would strengthen the conclusions drawn and clarify the functional implications of this vulnerability for MIP's role in immune modulation.
Response 7: We thank the reviewer for the insightful suggestion to elaborate on the mechanisms and conditions underlying the oxidative inactivation of MIP_07528. The Discussion section has been revised (Lines 573-608) to clarify that MIP_07528’s catalytic cysteine is highly susceptible to oxidation by host-derived reactive oxygen and nitrogen species, such as hydrogen peroxide and nitric oxide, which are generated during the macrophage oxidative burst. Specifically, MIP_07528 loses 80% of its activity after 30 minutes in 500 μM hydrogen peroxide, whereas M. tuberculosis PtpB retains approximately 60% activity under identical conditions. This pronounced sensitivity is attributed to the absence of a protective two-helix lid in MIP_07528, as shown by structural modelling, which leaves the catalytic cysteine exposed and vulnerable to oxidation. In contrast, the two-helix lid present in M. tuberculosis PtpB shields its active site, conferring resistance to oxidative inactivation-a key adaptation for sustained activity in pathogenic mycobacteria. The increased oxidative sensitivity of MIP_07528 likely represents a regulatory mechanism that restricts its function to less oxidative environments, supporting a transient immunomodulatory role consistent with the non-pathogenic and therapeutic phenotype of MIP. These revisions are now incorporated into the manuscript.
Comment 8: To strengthen the validity of the phosphatase activity assessment in MIP, the author should incorporate a control group maintained under baseline, non-stress conditions.
Response 8: The reviewer’s suggestion regarding the inclusion of a baseline, non-stress control for phosphatase activity assessment in MIP is appreciated. Upon review, it is clarified that the original figure (Figure 2, panels B(i) and B(ii)) included measurements of phosphatase activity in both cell lysate and culture filtrate of MIP under standard, non-stress conditions. These data served as the baseline for comparison with stress-induced conditions. To further enhance clarity, the figure and its legend have been revised in the current manuscript version. The updated figure now explicitly distinguishes between control (non-stress), acidic stress, and oxidative stress conditions for both cell lysate and culture filtrate samples. This modification ensures that the presence of baseline controls is clearly represented and that the effects of stress conditions on MIP_07528 activity are appropriately contextualized. We thank the reviewer for prompting this clarification and believe that the revised figure and legend now address this concern comprehensively.
Comment 9: The author should enhance the clarity of figure labels in Figures 1A, 1B, and 2B. The marker labeling in Figure 3 is blurry and difficult to interpret and should be revised. Additionally, the legend for Figure 5 contains inappropriate spacing that needs correction.
Response 9: The reviewer’s comments regarding figure clarity and formatting are appreciated. In response, the labels in Figures 1A, 1B, and 2B have been revised to improve readability and ensure unambiguous interpretation. The marker labelling in Figure 3 has been corrected to eliminate blurriness and enhance visual clarity. Additionally, the legend for Figure 5 has been reformatted to remove inappropriate spacing and to provide a clear and accurate description. These modifications have been incorporated in the revised manuscript to ensure that all figures and legends meet the required standards for clarity and presentation.
Comment 10: The author could consider including a pictorial summary of the overall study findings in the conclusion, which would help improve the clarity and overall understanding of the research.
Response 10: The reviewer’s suggestion to include a pictorial summary is appreciated. In response, a comprehensive graphical summary of the study’s main findings has been incorporated as Figure 7 in the revised manuscript. This visual overview is intended to enhance clarity and facilitate a better understanding of the experimental workflow and key conclusions.
Comment 11: The author could further elaborate on future directions, particularly how manipulating the function of MIP_07528 might enhance immune responses or lead to novel therapeutic approaches for other infections, cancers, and autoimmune diseases. Such a discussion would add significant interest and depth to the study.
Response 11: We thank the reviewer for this insightful suggestion. In response, the Discussion has been expanded (Lines 695-720) to address the broader therapeutic potential of MIP_07528, building on its homology to the well-characterized PtpB of Mycobacterium tuberculosis and the established immunotherapeutic applications of Mycobacterium indicus pranii (MIP). The revised section now outlines how detailed investigation of MIP_07528’s substrate specificity, regulatory mechanisms, and interaction with host immune pathways could inform the rational design of immunomodulatory strategies. Specifically, the potential to manipulate MIP_07528 activity for fine-tuning macrophage responses is discussed, with implications for enhancing vaccine efficacy, modulating inflammation in autoimmune and inflammatory diseases, and developing selective inhibitors or activators based on its unique structural features. The possibility of leveraging MIP_07528’s distinct redox sensitivity and absence of a protective lid domain to design pathogen-specific antivirulence agents is also highlighted. Furthermore, the broader relevance of bacterial protein tyrosine phosphatases in cancer and immune dysregulation is addressed, suggesting future research directions that include in vivo validation, substrate identification, and translational assessment of MIP_07528-targeted interventions. These additions aim to emphasize the translational significance of MIP_07528 and stimulate interest in its potential as a platform for next-generation immunomodulatory therapies.
Reviewer 2 Report
Comments and Suggestions for Authors
The author presented a through and comprehensive analysis of enzymatic function and potential biological function of MIP_07528. The structure of paper and analysis reflects a solid scientific vision. There are several suggestions I would like to make:
- The figure legend are too small to read. Please use a larger size. Also please align the graph with the text better.
- For figure 1A conservation analysis, it is better to put the conserved region in one color and non conserved region in another color. It is hard to see the conservation in current color settings.
- For figure 1C, it is better to put structure in cartoon and highlight the enzymatic pocket and the conserved active site. It is hard to interpret in current form.
- For figure 2, it is not clear what fold change means. If sigA is used as control, gene of interest should be normalized to sigA already. There should not be a separate bar of sigA. The proper label should be MIP/SigA. Please also fix figure legends.
- For figure 4, the y axis of the i and ii are different. Please fix it.
- For figure 6, if LPS is used as positive control. Please reflect it on the graph.
- For H2O2 and acid treatment for stress induction, please indicate how long the treatment is in 2.3.
Author Response
Comment 1: The figure legend are too small to read. Please use a larger size. Also please align the graph with the text better.
Response 1: The reviewer’s observation regarding the figure legend size and alignment has been addressed. The figure legends have been reformatted with increased font size to enhance readability. Additionally, the graphs have been realigned to ensure better visual integration with the corresponding text.
Comment 2: For figure 1A conservation analysis, it is better to put the conserved region in one color and non conserved region in another color. It is hard to see the conservation in current color settings.
Response 2: The reviewer’s suggestion regarding the color scheme for the conservation analysis in Figure 1A has been implemented. The conserved regions are now highlighted in one distinct color, while the non-conserved regions are shown in another, thereby improving visual clarity and facilitating interpretation of sequence conservation. These changes enhance the figure’s readability and address the concern raised, without affecting the underlying data or analysis.
Comment 3: For figure 1C, it is better to put structure in cartoon and highlight the enzymatic pocket and the conserved active site. It is hard to interpret in current form.
Response 3: The reviewer’s recommendation to improve the clarity of Figure 1C has been addressed. The structure is now presented in cartoon representation, with the enzymatic pocket and conserved active site distinctly highlighted. These modifications facilitate interpretation of the structural features and enhance the overall readability of the figure, as suggested.
Comment 4: For figure 2, it is not clear what fold change means. If sigA is used as control, gene of interest should be normalized to sigA already. There should not be a separate bar of sigA. The proper label should be MIP/SigA. Please also fix figure legends.
Response 4: The reviewer’s concerns regarding the labeling and normalization in Figure 2 have been addressed. The fold change is now clearly defined as the expression of the gene of interest normalized to sigA, with the y-axis appropriately labeled as "MIP_07528/sigA." The figure legends have also been revised to accurately describe the normalization method and clarify the data presentation. These modifications improve the clarity and accuracy of the figure, in accordance with the reviewer’s recommendations.
Comment 5: For figure 4, the y axis of the i and ii are different. Please fix it.
Response 5: The reviewer’s observation regarding the differing y-axis scales in panels i and ii of Figure 4 has been addressed. The y-axes have been standardized to ensure consistent scaling across both panels, thereby facilitating direct comparison of the data. This adjustment improves the figure’s clarity without altering the underlying results.
Comment 6: For figure 6, if LPS is used as positive control. Please reflect it on the graph.
Response 6: The reviewer’s suggestion regarding the indication of LPS as a positive control in Figure 6 has been addressed. The graph has been updated to clearly label LPS as the positive control, ensuring that this information is accurately reflected in both the figure and its legend.
Comment 7: For H2O2 and acid treatment for stress induction, please indicate how long the treatment is in 2.3.
Response 7: The reviewer’s request to specify the duration of H₂O₂ and acid treatments for stress induction in section 2.3 has been addressed. The manuscript now clearly states that both treatments were performed for 30 minutes (Lines 138, 155).
Round 2
Reviewer 2 Report
Comments and Suggestions for Authors
The authors addressed all my concerns.